# What Is in the Bank? Assessing Persistent Soil Seed Bank Density of *Sclerocactus wrightiae* (*Cactaceae*)

David Lariviere [1,*], Val Anderson [1], Robert Johnson [2] and Randy Larsen [1]

1   Department of Plant and Wildlife Sciences, Brigham Young University, 4105 LSB, Provo, UT 84602, USA; val_anderson@byu.edu (V.A.); randy_larsen@byu.edu (R.L.)
2   Department of Biology, Brigham Young University, 4102 LSB, Provo, UT 84602, USA; robert_johnson@byu.edu
*   Correspondence: david.d.lariviere@gmail.com

**Abstract:** Wright fishhook cactus is a small globose cactus endemic to an area of 280,000 ha in south-central Utah and was listed as endangered in October of 1979 by the U.S. Fish and Wildlife Service (USFWS). There is a general paucity of information about this species, and no published data on the seed bank for any species in the genus. Our objective with this study was to provide insight into the established seed bank density for this species. We processed 500 soil samples from various locations near individual cacti and potential neighboring nurse plants. We found that the species had a detectable seed bank of a size similar to other members of the Cactaceae family. Seed bank densities were the highest immediately adjacent to, and downslope from, parent plants. Our data indicate that areas within 20 cm of seed-producing cacti contain by far the greatest density of seeds. These areas should be given special consideration in future management plans for this species.

**Keywords:** cacti; seed bank; desert ecology; seed production; range management; endangered cacti; land management; globose cactus

## 1. Introduction

*Sclerocactus* is a genus of cactus described in 1922 by N.L. Britton and J.N. Rose typified with *Sclerocactus polyancistrus* [1]. Cacti in this genus are small and globose with hooked spines that curve inward. These distinctive spines are what have given the species its common name, "Wright fishhook cactus", due to their fishhook-like appearance. The range of this genus spans most of the American West, including the states of Arizona, California, Colorado, Nevada, New Mexico, Texas, and Utah. Northern Mexico also hosts a few species. There are 15 currently recognized species in this genus [2]. Many of these species are of limited range and consist of only a few small, scattered populations [3]. Of the fifteen described species, four are federally listed threatened and one endangered under the authority of the Endangered Species Act of 1973 in the United States [4]. Threatened species include the Pariette cactus (*Sclerocactus brevispinus*), Colorado hookless cactus (*Sclerocactus glaucus*), Mesa Verde cactus (*Sclerocactus mesa-verdae*) and Uintah Basin hookless cactus (*Sclerocactus wetlandicus*). Of particular conservation interest is the only species in the genus listed as endangered, the Wright fishhook cactus (*Sclerocactus wrightiae*) [4]. This species is endemic to three counties in the south-central region of Utah, USA [5,6] (Figure 1).

In 1966, Wright fishhook cactus was distinguished from the much more widely spread little-flower fishhook cactus (*Sclerocactus parviflorus*) by its white flower petals and magenta filaments [5]. Wright fishhook cactus has the ability to retract into the soil during dormancy, emerging with sufficient rainfall to bloom in late April to early May, making them uniquely difficult to inventory in drought years [2]. In 1979, Wright Fishhook cactus was listed as endangered by the U.S. Fish and Wildlife Service (USFWS), citing its very limited range, population size, and the prevalence of poaching by international cactus hobbyists [7]. Ancillary mortality factors include predation by the opuntia borer-beetle, blister beetle, and

Ord kangaroo rat [8]. These mortality factors primarily impact reproductively mature, adult individuals. Various other threats have also been identified, including climate change and oil and gas development [8–10]. However, there has been limited research directly assessing the effects of these other factors. Negative impacts from cattle traffic were previously considered a concern; however, recent research does not support that supposition [11,12].

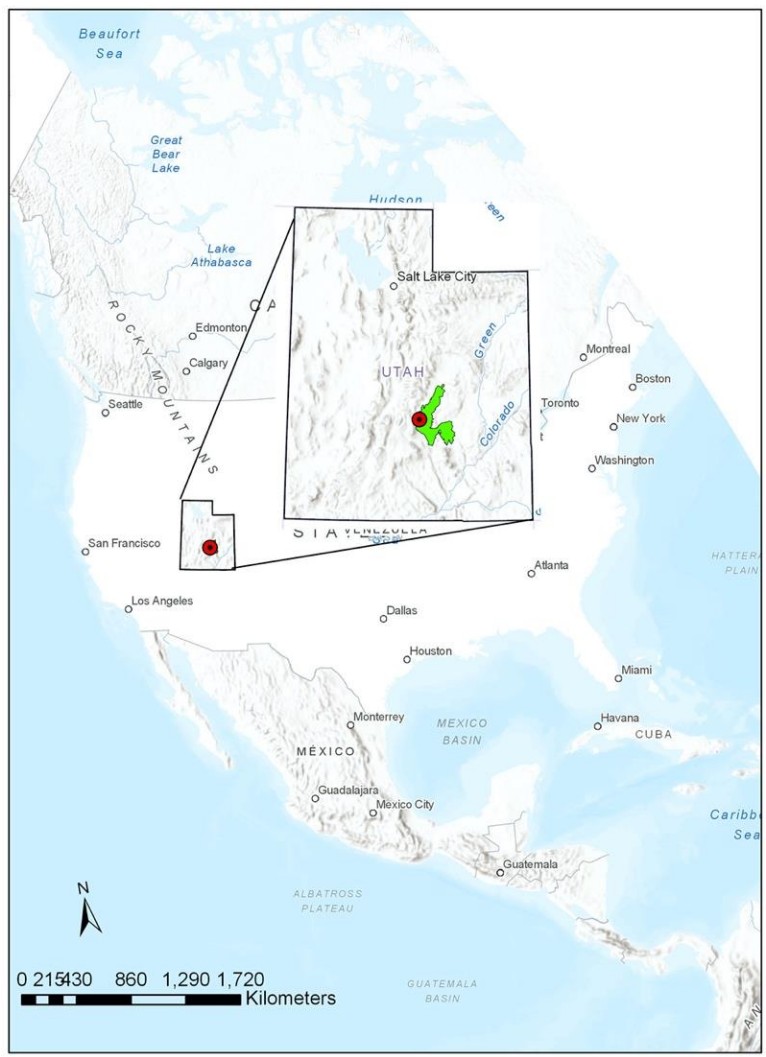

**Figure 1.** Range map of *Sclerocactus wrightiae* and the red point is the study site. Centroid of range: 38.5318460, −111.2391515.

A well-established soil seed bank is critical for many species, especially in areas where seed mortality through predation is high [13]. Globose cacti have been shown to have medium-term, persistent seed banks that have been shown to be viable for up to 3–7 years [14,15]. In contrast, seeds from a columnar cactus, *Stenocereus stellatus*, have been shown to be viable only during a window between 6 and 10 months after leaving their parent plant [16], while other species of the columnar cactus have no detectable seed bank at all [17]. Within Cactaceae, seed bank size appears to vary widely between genus and even species [18–20].

It has been found that within Cactaceae, seeds are often either clustered near the base of the parent plant or found under nearby nurse plants [16,20,21]. A nurse plant can loosely be defined as a plant that provides a favorable microhabitat, or protection from predation, for seeds. Additionally, the presence of a persistent seed bank within family Cactaceae has been shown to have large spatiotemporal demographic distributions within populations [22]. A persistent seed bank describes seeds that have been in the soil for at

least one calendar year. The objective of our study was to determine if *Sclerocactus wrightiae* has a detectable persistent seed bank of similar size to other members of Cactaceae, and further, whether that seed bank exists primarily near parent plants or nurse plants. It should be noted that the size of the transient seedbank for this species was not examined in this study. The number of seeds in a transient seed bank has been shown to be greater than those in a persistent seed bank within Cactaceae [19].

In the genus *Sclerocactus*, germination rates of 75% from the seed bank have been shown to be possible in a controlled greenhouse setting [23]. Under typical natural conditions, germination rates for the genus have been hypothesized to be much lower [24]. However, no work has been published assessing *Sclerocactus* seed germination or viability under natural conditions. Regeneration and recruitment rates from seed for this genus are thus currently unknown. Further, the age of seeds at time of germination is also not currently known. Importantly, the high variability in the spatiotemporal demographic distribution of persistent seedbank observed in other members of Cactaceae should be taken into consideration when estimating recruitment rates for this cacti species. More broadly however, the very existence of a detectable seed bank for Wright fishhook cactus and the *Sclerocactus* genus at large has never been examined. Our hypothesis was that Wright Fishhook cactus has a detectable persistent seed bank that is similar in size to other members of Cactaceae.

## 2. Materials and Methods

### 2.1. Study Site

The study site is located on four hectares of private property approximately 14 km south of Fremont Junction, Utah (latitude 38°63′ N, longitude 111°33′ W), in what is known as the Last Chance Wash. The community structure is that of desert scrub steppe. This is a desert grassland fringe community that commonly surrounds major desert complexes [25]. The region has an arid climate, with an average annual precipitation of 190 mm [26]. The soil at the site is sandy clay loam in texture and is underlain by alluvium [27]. The dominant native plant species in this desert grassland fringe community are as follows: shadscale (*Atriplex confertifolia*), sand buckwheat (*Eriogonum leptocladon*), alkali sacaton (*Sporobolus airoides*), galleta (*Hilaria jamesii*), Torrey's ephedra (*Ephedra torreyana*), four-wing saltbush (*Atriplex canescens*), Indian rice grass (*Achnatherum hymenoides*), and prickly pears (*Opuntia* sp.). Non-native species frequently found in this community include Russian thistle (*Salsola tragus*) and halogeton (*Halogeton glomeratus*).

### 2.2. Soil Sampling

Although seed bank assessments have never been completed for any member of the genus *Sclerocactus*, seed bank densities of other species within Cactaceae were considered during experimental design relative to potential methodology as well as reasonable expectations of estimating the density for this species [16–18,20,28]. Using results from these studies, we expected to find between one and two percent of seeds escaping predation on our study site. Reproductively mature *Sclerocactus wrightiae* have been shown to produce an average of 91 seeds per year [12]. Our study area has a population size estimated to be approximately 500 reproductively mature individuals. Given these figures, we estimated the total population seed rain for this site at approximately 45,500 seeds per year. Using the above estimate of one to two percent of seeds making it into the seed bank each year, we would expect between 455 and 910 seeds to escape predation in a given year for the entirety of this population.

We surveyed the study site for Wright Fishhook cactus in June 2021, and 50 reproductively mature cacti were selected for use in the study. Reproductively mature individuals were previously defined by Ronald Kass [29] as cacti within size class two (2.1–4 cm diameter), and size class three (4.1–9 cm diameter). Kass also denoted a fourth size class (>9 cm), but no size class four individuals were located on the study site during our survey or during previous surveys [12]. Because size class three cacti were the primary reproductive size

found in the survey, we limited our study sample to this size class. This limitation is more representative of the larger Wright fishhook cactus extant population, as demographically, among reproductively mature individuals, size class 3 is the most common [29].

In order to assess the size and location of the seed bank for this species, we surveyed seed density in the soil by extracting 500 samples, each consisting of 105 cubic centimeters of soil. Soil was sampled with a 7 cm long by 5 cm wide scoop at 3 cm depths. Due to the length of the sampled area being 7 cm and the diameter of a reproductively mature, size class three Wright fishhook cactus averaging 7 cm [29], our sample area surveyed the entire downslope aspect of an individual.

Sample areas were divided into three locations: downslope of a cactus, upslope of a cactus, and at potential nurse plants in the interspace between individuals. Potential nurse plant sites were comprised of four species, alkali sacaton (*Sporobolus airoides*), galleta (*Hilaria jamesii*), Indian rice grass (*Achnatherum hymenoides*), and prickly pear (*Opuntia* sp.). These sites were selected to provide a clear picture of where the seed bank was primarily located. There were 50 downslope, 10 upslope and 10 nurse plant locations sampled. Upslope and downslope samples were taken from the same cacti. Downslope locations were considered the most likely to harbor seeds as seed dispersion through rainwater has been shown to be a significant factor in the movement and dispersal of seeds, especially in arid and semi-arid ecosystems [30,31]. Due to the location of nearby mountains and the canyon topography caused by Last Chance creek, the downslope side of these cacti was also the predominate leeward area, likely furthering seed accumulation (Figure 2). Due to time and budget constraints, these downslope sites were selected in greater numbers than the upslope locations.

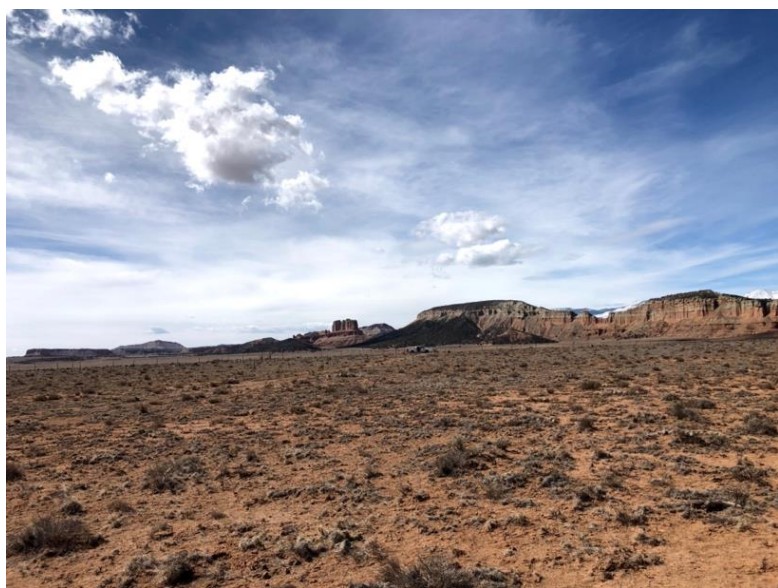

**Figure 2.** Photograph of the study site.

At our study site, the slopes range from approximately 3 degrees to as much as 30 degrees immediately adjacent to cacti. However, the greater slopes are generally not sustained for more than six inches before undulating into a different slope or aspect. The overall topography of the area is relatively flat, with perennial bunchgrasses and forbs forming pedestal-like raised areas (Figure 2). While there is a slight consistent slope from the base of hills to the valley bottoms, there was no consistently significant downslope aspect among all cacti. Individual cacti are almost always located on top of small pedestals with obvious upslope and downslope aspects (Figure 3).

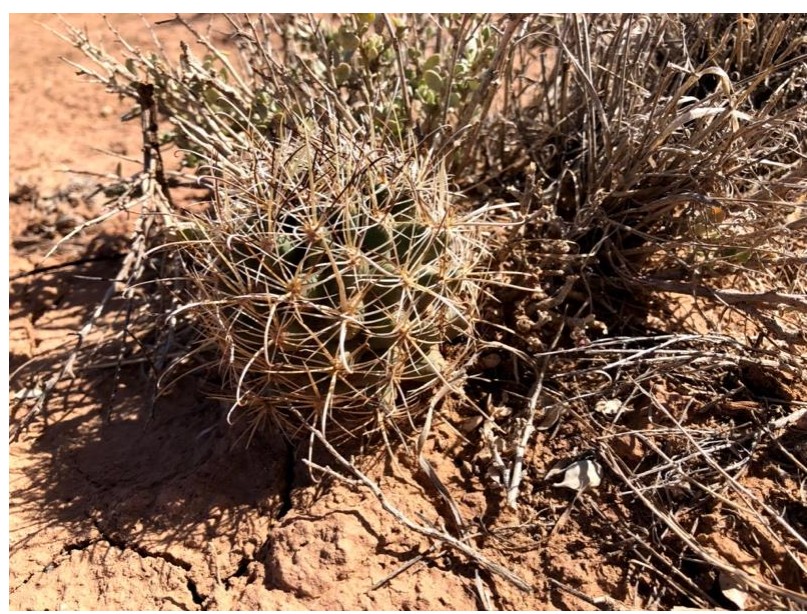

**Figure 3.** Cacti in habitat.

These upslope and downslope locations were sub-divided into distance and depth from the parent plant. Radiating away from a cactus, distances of 0 cm–5 cm, 5 cm–10 cm, 10 cm–15 cm, and 15 cm–20 cm were sampled. Each sample extracted all soil from the soil surface to a vertical depth of 3 cm. Following this, these same locations were sampled again at a soil depth from 3 cm–6 cm.

A total of 400 individual downslope, and 80 individual upslope samples were taken across all cacti sampled. Due to the extremely low expected density of the seed bank, we focused our sampling to within 20 cm of the base of reproductively mature cacti. To ensure that all seeds encountered were from prior years, sampling was conducted prior to the current season seed drop. As such, all seeds discovered were at least one year old, had escaped predation and transitioned into the seed bank.

Ten nurse plant locations were randomly selected and sampled with the same 7 cm by 5 cm wide scoop at a surface level down to 3 cm, and at the second soil depth of 3 cm to 6 cm below the soil surface. There were 20 individual samples taken. Samples were taken on the downslope and leeward side of nurse plants in order to best replicate the sampling procedure of the parent plants. Randomization was accomplished through the use of a random number generator dictating the cardinal direction and number of paces to take from a previously sampled cactus. No random location was closer than three meters to a Wright fishhook cactus individual.

Soil samples were taken to the lab where they were sieved and searched for the presence of Wright fishhook cactus seeds. There were two sieve sizes used. The first was large, meant only to remove stones. The second had 1 mm openings that prevented the seeds from passing through. Prior to sieve use, 40 random Wright Fishhook cactus seeds collected during a previous study were measured for size. The minimum seed was found to be 2.86 mm, and the largest was found to be 3.44 mm. The mean length was 3.17 mm with a standard error of 0.02 mm. A typical seed was photographed (Figure 4). Water was poured over samples in the second sieve in order to break apart any compacted soil and allow its passage through.

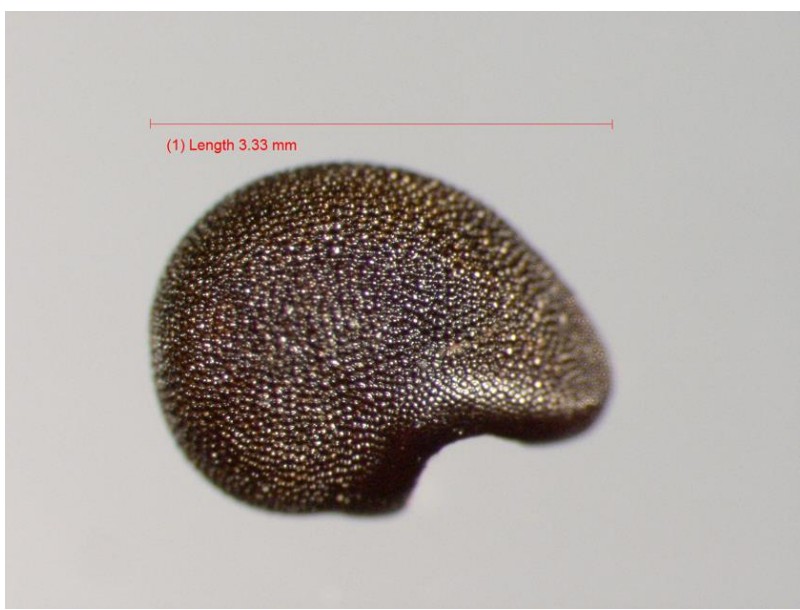

**Figure 4.** Wright fishhook cactus seed.

The number of seeds in each sample was recorded. Due to their distinct coloration and shape, seeds from this species were readily distinguished from the more common grass (Gramineae) seeds found in the samples. Use of a microscope was not necessary during seed identification.

After identification, a viability assessment was conducted. All recovered seeds were pricked and soaked in water for 16 h at 25 °C. Afterwards, the seeds were soaked in a 1% tetrazolium (2, 3, 5-triphenyl tetrazolium chloride) solution [32] for 20 h at 25 °C, then dissected under a stereo microscope to assess staining for viability. Of the recovered seeds, 32% proved to be viable. It should be noted again that the seeds sampled solely represent the persistent seedbank for this species. It was not possible to age the recovered seeds, but all had spent at least one calendar year in the soil.

*2.3. Analysis*

To evaluate any seed density differences between upslope, downslope, and nurse plant locations, while also checking for model accuracy, we used a generalized, linear, mixed-effects model with a binomial distribution for error structure within the "lme4" package in program R [33–35].

Seed bank location (upslope, downslope, or nurse plant), distance from the parent plant (in cm), and depth of the sample (in cm) were identified as potential explanatory variables. We set unique ID as a random effect and included it in all models. Because our analysis was observational in nature, we used an information–theoretical approach and model selection [36,37]. We first created a set of eight a priori models with combinations of the above listed explanatory variables to represent hypotheses about which factors influence prediction accuracy. Before creating our models, we evaluated the potential for multicollinearity within our dataset for explanatory variables using Pearson's correlation coefficient for continuous variables and did not include any variables with an $|r| > 0.60$ in the same model. Following model selection, we further evaluated the potential for multicollinearity using variation inflation factors (VIF) and a cutoff of 10 [33,38]. We ranked a priori models using Akaike's Information Criteria adjusted for small sample sizes and AICc weights [36,37]. We further assessed our ranked list of models for any evidence of uninformative parameters; none were found [39,40].

### 3. Results

Across 500 individual samples, a total of 43 seeds were found. Within the sample area, seed density was the highest immediately adjacent and downslope to the parent plant at both soil depths. A burial depth of 0–3 cm yielded greater densities of seeds than the deeper substrate layer (3–6 cm depth). Most seeds were found within 5 cm of the parent plant with fewer found between 5 and 10 cm, between 10 and 15 cm, and 15 to 20 cm (Figure 5). As expected, depth distribution favored the shallow depth with 56% of the seeds found at a soil depth between 0 and 3 cm, while 44% were found between 3cm and 6cm. Seeds were most often discovered downslope (98% of the total), while 2% were found upslope of parent plants, and 0% in nurse plant samples.

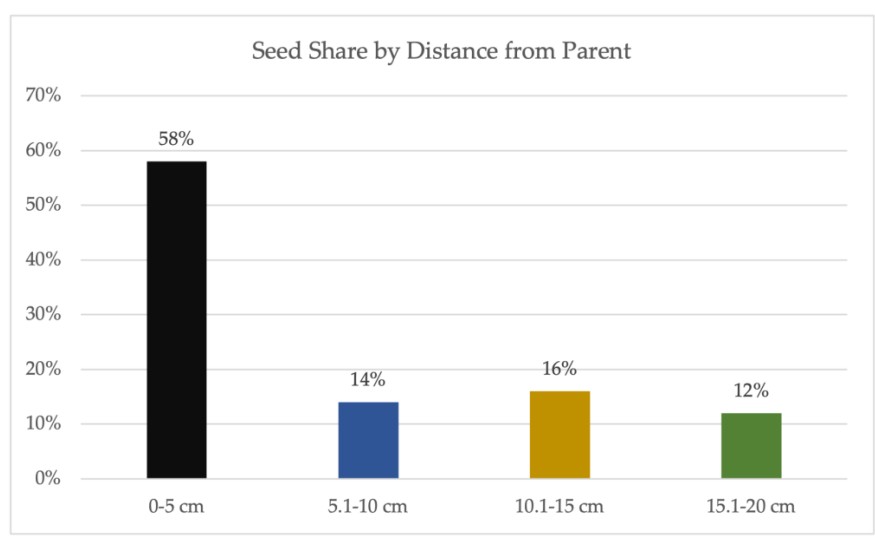

**Figure 5.** Bar graph of seed share by distance from the parent plant.

Models including the distance of the sample from the nearest parent plant and sample depth carried the most AICc weight and these explanatory variables occurred in all top-ranked models (Table 1). We found less support for depth in our modelling effort as it did not occur in the most-supported model (Table 1). None of the models contained uninformative parameters. Variance inflation factors for supported models were low (1.025–1.032), indicating there was limited correlation between the explanatory variables.

**Table 1.** Model selection table for 8 a priori models of seed density estimates for *Sclerocactus wrightiae* sampled in 2021 in Utah, USA.

| Model Structure | df [a] | logLik [b] | AICc [c] | Delta [d] | Weight [e] |
|---|---|---|---|---|---|
| Presence~Position + Distance | 4 | −106.93 | 221.9 | 0.00 | 0.395 |
| Presence~Depth + Position + Distance | 5 | −106.64 | 223.4 | 1.46 | 0.191 |
| Presence~Distance | 2 | −109.77 | 223.6 | 1.63 | 0.175 |
| Presence~Position | 3 | −109.40 | 224.8 | 2.91 | 0.092 |
| Presence~Depth + Distance | 3 | −109.48 | 225.0 | 3.08 | 0.085 |
| Presence~Depth + Position | 4 | −109.11 | 226.3 | 4.36 | 0.045 |
| Presence~1 | 1 | −113.48 | 229.0 | 7.03 | 0.012 |
| Presence~Depth | 2 | −113.20 | 230.4 | 8.48 | 0.006 |

[a] Degrees of freedom. [b] Log likelihood. [c] Akaike's Information Criteria. [d] AIC relative to the best fitting model. [e] Akaike weight.

Distance from the seed-producing plant significantly influenced the quantity of seeds present in a sample. The position of sample location (upslope, downslope, nurse plant) also significantly influenced the number of seeds present. Sample depth did influence model quality and was a variable in our second-ranked model (Table 1); however, 85% confidence intervals around the beta estimate overlapped zero (Table 2), suggesting it was not as influential as other variables.

**Table 2.** Model-averaged estimates with an 85% confidence interval for presence of Wright fishhook cactus seeds in relation to position, distance, and depth. Data collected in 2021 in Utah, USA.

|  | Estimate [a] | Std. Error [b] | Adj. SE [c] | z-Value [d] | LCI [e] | UCI [f] |
|---|---|---|---|---|---|---|
| Intercept | −2.1264 | 0.3437 | 0.3443 | 6.177 | −2.6217 | −1.6312 |
| Position (Nurse plant) | 3.3989 | 884.2836 | 886.4470 | 0.004 | −1271.530 | 1278.3277 |
| Position (Upslope) | −1.8297 | 1.0258 | 1.0283 | 1.779 | −3.3086 | −0.3508 |
| Distance | −0.0772 | 0.0358 | 0.03586 | 2.154 | −0.1288 | −0.0257 |
| Depth | −0.2905 | 0.3823 | 0.3842 | 0.756 | −0.8431 | 0.2620 |

[a] Model-averaged β estimate. [b] Standard error. [c] Adjusted R-squared. [d] Number of standard deviations from mean. [e] Lower confidence interval. [f] Upper confidence interval.

## 4. Discussion

We hypothesized that Wright fishhook cactus would have a detectable seed bank of similar proportion to the total seeds produced compared to the other members of Cactaceae. We found a total of 43 seeds across the 100 individual parent plants sampled. This equates to a mean of slightly fewer than one (0.86) seed per parent plant within the area sampled. We sampled a surface area of 280 square centimeters within a 20 cm radius of the base of parent plants. The area we sampled represents 17.5% of the total surface area within this 20 cm radius. If one were to extrapolate our findings, they might expect to find a total of approximately five cacti seeds within a 20 cm radius of a parent plant at soil depths down to 6 cm. Again, this number would represent the persistent seed bank for an individual parent plant, rather than the transient seed bank for any given year.

With seed-rain loss in other members of Cactaceae typically near 98–99% [16–18,20,21], and mean seed production per reproductive size individual for this location being 91 seeds [12], we estimated that approximately 1 seed per year would make it into the seed bank for a given individual in this population. The leading cause for the relatively high seed loss has been shown to be granivory [16–18,20,21,41].

We observed ants (*Formicidae* sp.) harvesting seeds both directly from the cacti before dispersal as well as collecting seed from the soil surface. During the days prior to the dehiscing of cacti fruiting bodies, when these fruiting bodies have deteriorated to the point of nearly allowing the first seeds to escape, ants will actively chew through these bodies and harvest the seeds within. Ants are able to strip nearly every seed from a cactus over the course of a single day. Additionally, personal observation of several of the authors, as well as by BLM field personnel, has confirmed lagomorphs (Lagomorpha) consuming Wright fishhook cactus, which may directly or indirectly eliminate the fruiting bodies of affected cactus. The majority of such observations occurred in early spring, as the cacti began to take up water and swell in size, but before other vegetation had achieved significant growth for the year. Given these observations, we expected seed predation rates for this species to closely align with the 98–99% observed in other members of Cactaceae as noted above.

The age of the seed bank, and therefore the total number of seeds we would have expected to find given the 98–99% seed rain loss was not discernable. While the larger size classes of this species denote reproductive maturity and "adult" status [5], there currently exists no standard procedure for ascertaining the age in years from individual specimens within this species. Doing so could have provided an indication of the number of years of seed rain a sampled area had experienced. There is also currently no method for aging the seeds themselves.

Across *Cactaceae*, the presence of sheltering microhabitats such as nurse plants or soil surface cracks lead to greater seed bank densities and recruitment rates primarily due to their sheltering of seeds from predators [15,17,41,42]. Additionally, the slight shading of cacti seedlings during the early stages of germination in arid environments has been shown to increase recruitment rates [43]. Given the relative openness and paucity of vertical vegetation structure of this desert grassland fringe terrain, the parent plant itself may often serve as the best means of shelter from seed predators and provide the most suitable microhabitat during the early stages of germination. While no work has been carried out

assessing seed germination rates for this species in different microhabitats, the authors have observed numerous cacti of the smallest size class growing immediately adjacent to parent plants. However, it is not clear whether these individuals come from seeds. If they do, it would be in line with our finding that this species has no detectable seed bank at potential nurse plant sites greater than three meters from an individual cactus. Future studies should consider decreasing the minimum distance from the parent plant to the potential nurse plant site to less than three meters due to our finding of low seed bank density. Nurse plants nearer the parent pant may yield more seeds in the persistent seed bank.

The lack of any seeds being present in samples collected at nurse plant sites should not be taken to indicate that this species does not benefit from such locations. The limited size of the seed bank as well as the plethora of potential nurse plant sites should rather be evidence of the incredibly low seed density per unit area beyond 20 cm from the base of a parent plant. The authors of this paper have personally observed numerous Wright fishhook cacti germinating in favorable micro-habitats far from potential parent plants in places such as soil cracks, bunch grass pedestals, and cattle hoof prints.

Germination rates for this genus have been found to reach 75% under controlled greenhouse settings [24] but have been hypothesized to be far lower under natural conditions, possibly as low as 2% [25]. While our study did not examine germination rates of the existing seed bank, these studies draw attention to the importance of a robust seed bank for this species. Seedling recruitment over the past decade has been high across the range for this species, further emphasizing the importance of a healthy seed bank [44].

The nature of potential dispersal methods for seeds of this species makes seed bank density calculations challenging. Wind, rainwater, and carriage by animals such as ants and lagomorphs are all feasible dispersal vectors for these seeds. The relative isolation of populations of Wright fishhook cactus from each other implies that seed dispersal can, and has, reached miles from the parent plant.

We found no detectable seed bank greater than 20 cm from the base of a cactus. However, our data should not be taken to conclude that no seed bank exists for this species beyond 20 cm from a parent plant. Rather, seed bank densities beyond this range simply become sufficiently low as to not be detectable with our approach. This is obvious due to new populations of the species forming far from existing ones.

A small but detectable seed bank does exist for this species within a 20 cm radius of the base of parent plants. Our data indicate that approximately five seeds within this radius for a given parent plant is typical. While not a high number, this seed bank is apparently sufficient to sustain populations over time and for new populations to develop. However, future management for this species should note the apparently high seed predation rate for this species, and the importance of the seeds that do avoid this pressure and transition into the seed bank. These seeds, given their limited number, appear to be crucial to the long-term viability of a population.

Our study was limited by observations and data collection on a single population at one site (private land holding) and thus its ability to make larger predictions about seed bank density of this species more broadly. Future research should consider multiple populations of this species spanning the entirety of its range. Understanding how many years of seed-rain our samples represented would provide a far clearer picture as to annual seed loss. While germination rates have been studied for this genus in a greenhouse setting [23], species-specific work could be beneficial to more thoroughly understanding seed bank recruitment in a natural setting. Further, rather than relying on observation, an experimental in situ seed burial study for this species would be useful. In situ seed germination rates for cactaceae have been shown to be far lower than ex situ seed storage [45]. These limitations taken together provide a strong case for further research on this topic in order to demonstrate with greater significance what role the seed bank of this species plays in annual recruitment.

**Author Contributions:** Conceptualization, D.L., V.A. and R.J.; methodology, D.L., V.A. and R.J.; software, D.L. and R.L.; validation, R.L.; formal analysis, R.L.; investigation, D.L.; resources, D.L., V.A. and R.J.; data curation, D.L.; writing—original draft preparation, D.L.; writing—review and editing, V.A., R.L. and R.J.; visualization, D.L. and R.L.; supervision, V.A.; project administration, V.A.; funding, V.A. All authors have read and agreed to the published version of the manuscript.

**Funding:** This research was funded by the Bureau of Land Management Grant Number: DOI BLM L18AC00042.

**Institutional Review Board Statement:** No voucher specimens of this endangered species (*Sclerocactus wrightiae*) were collected in order to preserve the study population.

**Data Availability Statement:** The data presented in this study are publicly available in figshare at 10.6084/m9.figshare.24282259.

**Conflicts of Interest:** The authors declare no conflicts of interest. The funders had no role in the design of the study; in the collection, analyses, or interpretation of data; in the writing of the manuscript; or in the decision to publish the results. Prior work by the authors published as "An Assessment of Cattle Traffic on, and Seed Dispersion Patterns of, *Sclerocactus wrightiae* of, *Sclerocactus wrightiae*" published in this journal has been referenced where relevant but is otherwise distinct from and created no conflicts of interest for this article.

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
