# Peer review of "What Is in the Bank? Assessing Persistent Soil Seed Bank Density of Sclerocactus wrightiae (Cactaceae)"

_diversity, doi:10.3390/d16030133_

Round 1
Reviewer 1 Report
Comments and Suggestions for Authors
Line 2: Add ‘soil’ to make it ‘Assessing soil seed bank density…’
Line 3: Add ‘(Cactaceae)’ after genus and species
Line 13: Delete ‘in general’
Line 16: Make ‘seedbank’ 2 words – here and throughout (many times)
Line 25: ‘fishhook’ and not ‘foshhook’
Line 31: Should ‘endangered species act’ be capitalised? And include a year?
Line 35: Should ‘endangered’ be capitalised?
Line 39: Italicise genus and species in the figure legend. Remove parentheses.
Line 47: again, should ‘endangered’ be capitalised?
Line 56: Upon first mention, add ‘soil’ to make it ‘A well-established soil seed bank is critical…’. To avoid confusion with ex situ conservation seed banks.
Line 62: add ‘to’ to make it ‘seed bank size appears to vary widely..’
Line 67: Delete space before ‘Using results from…’
Line 69: correct spelling of ‘Reproductivley’ and italisise Sclerocactus wrightiae
Lines 64-86: The approach (including experimental design) and hypotheses are currently somewhat mixed up in the two final paragraphs of the Introduction. Suggest moving experimental design to the Materials and Methods section, and clearly stating the aims of this study prior to the hypotheses.
I also suggest a separate prior paragraph that pulls together everything known/not known about a) the regeneration of S. wrightiae (and related species) via seed and b) the role of the soil seed bank. Include what a nurse plant is and its perceived/documented role.
Also, introduce the difference between ‘persistent’ and ‘transient’ soil seed banks, and make clear that you were aiming to investigate the persistent soil seed bank.
Line 102: Suggest this subheading changed to ‘Soil sampling’, and putting all sampling methodology together here to avoid repetition.
Line 122: Can you explain why an unequal number of each site type (downslope, upslope and nursery plant) were sampled?
Line 153-156: These methods seem scant given the importance. Can you provide more info on seed size and shape in the Intro? Was a microscope needed to find seeds in the soil? Most soil seed bank studies adopt the germination method, so your approach needs justification.
Line 178-187: Consider creating bar or pie charts to visually present some portion of these results (just a thought)
Line 244: add ‘to’ to make it ‘Ants are able to strip…’
Line 246: define ‘lagomorphs’
Line 268: change ‘have’ to ‘has’ to make it ‘the light shading… has been shown…’
Line 288: change ‘dispersion’ to ‘dispersal’
Line 309: how would you have determined the age the seeds?
Line 308: ‘Future research’ - Consider proposing artificial seed ageing experiments and in situ seed burial experiments, with references. Consider proposing that you test seeds for viability (germination trials, X-ray analysis and/or TZ testing), with refs.
Author Response
All grammar and word change suggestions were incorporated.
Lines 64-86: I reorganized based on your feedback so that the discussion about experimental design falls under the methods section. Further, I included the paragraph you suggested and added more details throughout the paper on persistent vs transient seeds.
Line 122: I offered further clarification on why a reduced number of upslope and nurse plant sites were selected. This was primarily driven by a lack of funding for student employees to sift the nearly 1000 additional soil samples sampling more of these sites would have entailed.
Line 153-156: I have added significant information to this section, and included the previously omitted results of the seed viability assessment we carried out.
Line 178-187: I have included a pie chart for better visualization
Line 308: I added a discussion on the value of in situ and ex situ work for this species.

Reviewer 2 Report
Comments and Suggestions for Authors General comments: All manuscript sections, in general, are well written. Introduction: Information of the study species is mentioned and relevance of the study present in this section but an objective is needed. Material and Methods: This section needs some clarifications. Results: They are well exposed in the text. Discussion: Authors discussed results. A conclusion about the importance of the seed bank in relation to the state of conservation of the species is lacking.Specific comments:
Abstract:
The objective of the work needs to be mentioned. Also in the introduction, before the hypothesis.
Add one or two sentences for discussion and conclusions. What importance does the seed bank have for this species and its conservation?
Figure 1.Line 39: Eliminate the parenthesis.
Name of the species in italic. Review throughout the manuscript
Line 61. Seed bank. Review throughout the manuscript
Line 85-86: To discuss the proposed hypothesis, it should be noted that the authors sampled the seed bank before seed dispersal. They would likely have found more seeds if sampled post-dispersal. When is the peak germination pulse of seeds, immediately after dispersal?
M&M
Line 102: Maybe you can select another more appropriate subtitle: Sampling design?
Line 114: Not all samples have the same depth, so they do not have the same volume.
Line 120: What is the composition of nurse plants? Shrubs? Grasses?
Line 122: nurse plant (without the hyphen). Review throughout the manuscript
Line 122: Did you take the soil sample at upslope and downslope of the same cacti individual?
What is the predominant wind direction? This can also affect seed distribution, with a greater accumulation occurring leeward of the parent plants.
Line 147: Did the samples taken from the same side of plant nurses? Eg. leeward, windward? Where were the samples extracted?
7cm long?
Line 153: Did you use a method to test the seed viability of seeds (e.g. tretrazolium)? Did you know this information from other Cactus congeners?
Please include size and shape information of seeds. Also the size of mesh used to sieve the soil samples.
Results
Line 178: 43 seeds in a total volume of 52500 cm3. This value does not match what is mentioned in line 299 (43 seeds/m2). Review.
Discussion
Line 230: This statement doesn't make sense. Less than one seed per parent plant?
Line 263: How did your plan to determine the age of the found seeds?
Line 270: Have you recorded seed germination near the parent plant?
Line 272: Three meters seems like a considerable distance for a seed to be dispersed. Perhaps sampling nurse plants closer to cactus plants would have yielded a different result.
Line 280: What would be those favorable microsites?
Translate: It would be good to mention the importance of the seed bank given the conservation status of the species.

Author Response
All grammar and spelling comments and suggestions have been incorporated. Further, where new titles or labels were suggested, these were updated. An objective was added to both the abstract and introduction.
Line 85-86: The scope of this study was to examine the persistent seed bank for this species. The paper has been updated to reflect this focus better.
Line 114: All samples were the same depth. I have updated the wording to make this more clear in the paper.
Line 120: I have updated to include the nurse plant species
Line 147: I have updated the sample location descriptions to include leeward/windward information
Line 153: Updated to include the results of our seed viability assessment. Updated to include seed size. Also included a photo of a typical seed.
Line 178: Corrected the number of expected seeds per square meter to reflect our findings. (original number was an oversight and restatement of total seeds recovered from our samples)
Line 230: The average seeds recovered per plant has been updated to be more clear.
Line 263: We did not plan on aging the seeds. Saying as much was an oversight.
Line 270: Unfortunately germination work outside of a greenhouse setting has never been done for this genus. I have heard that a researcher under a grant from Capitol Reef National Park is currently collecting data on this topic.
Line 272: In hindsight, a shorter distance liklely would have been better. I have included that in the discussion.
Line 280: a description and examples have been included

Reviewer 3 Report
Comments and Suggestions for Authors
This is a valuable research work, and I have only three small suggestions:
1. Too many keywords
2. It is recommended to add more field survey photos in Figures 2 and 3.
3. References should be checked and corrected according to the journal format.
Author Response
- Number of keywords has been reduced.
- more photos have been added
- References have been updated and standardized

Round 2
Reviewer 2 Report
Comments and Suggestions for Authors
This version has improved, and the authors have satisfactorily addressed my comments. I believe that the manuscript is ready for publication. Congratulations. I have some final suggestions.
Figure 2 and 3. I think these figures can be combined into one, indicating that they represent the study area. Perhaps provide a brief description of what is intended to be conveyed. I suggest the same for Figures 3 and 4.
Figure 5.I believe this figure can be improved. You could create a bar chart with different colors to represent these results. Eliminate the percentage values in the text, as they are redundant.
Line 746: This sentence is not clear. One option could be: It is not clear whether these individuals come from seeds.
Line 747: Independently
Line 864: Sufficient
Line 871: Broadly?
Author Response
Spelling and grammar have been updated.
Figures 2/3 and 4/5 were added at the request of another reviewer. The authors agree that they are superfluous.
Figure 5 has been updated
line 317: Wording has been updated for clarity
